# Detection of *Anaplasma phagocytophilum* DNA in Deer Keds: Massachusetts, USA

**DOI:** 10.3390/insects16010042

**Published:** 2025-01-04

**Authors:** Patrick Pearson, Guang Xu, Eric L. Siegel, Mileena Ryan, Connor Rich, Martin J. R. Feehan, Blake Dinius, Shaun M. McAuliffe, Patrick Roden-Reynolds, Stephen M. Rich

**Affiliations:** 1Laboratory of Medical Zoology, Department of Microbiology, University of Massachusetts, Amherst, MA 01003, USA; pbpearson@umass.edu (P.P.); gxu@umass.edu (G.X.); esiegel@umass.edu (E.L.S.); mileenaryan@umass.edu (M.R.); crich@umass.edu (C.R.); 2New England Center of Excellence in Vector-Borne Disease, University of Massachusetts, Amherst, MA 01003, USA; martin.feehan@mass.gov; 3Massachusetts Division of Fisheries and Wildlife, Westborough, MA 01581, USA; 4Department of Natural Resources and the Environment, Cornell University, Ithaca, NY 14853, USA; 5Plymouth County Extension, Plymouth, MA 02360, USA; bdinius@plymouthcountyma.gov; 6Hopkinton Health Department, Hopkinton, MA 01748, USA; smcauliffe@hopkintonma.gov; 7Martha’s Vineyard Tick-Borne Illness Reduction Initiative, Edgartown, MA 02539, USA; biologist@dukescounty.org

**Keywords:** deer keds, *Lipoptena*, white-tailed deer, *Odocoileus virginianus*, *Anaplasma*, ectoparasites, pathogen prevalence

## Abstract

Deer keds are parasitic flies (Diptera) that feed on deer. They occasionally bite humans, resulting in localized skin irritation. There are many knowledge gaps persisting around the distribution of deer keds and the pathogenic microorganisms that they may transmit. In this study, 99 deer keds were collected from white-tailed deer across Massachusetts (northeastern United States). These deer keds were screened for the presence of six tick-borne pathogens using real-time PCR testing. A tick-borne bacterium, *Anaplasma phagocytophilum*, was found in almost 1/3 of the tested deer keds. Because of host association patterns of defined *A. phagocytophilum* strains, these are most likely not pathogenic to humans. No other pathogens tested were identified. These results suggest that the risk of the transmission of pathogens to humans by deer keds may be low. Based on the feeding behavior of deer keds (feeding on only one host compared to some ectoparasites that may require blood meals from multiple hosts), deer keds may be a useful tool for the monitoring and surveillance of pathogens associated with deer. Future work is needed to further assess the potential significance of deer keds to public, wildlife, and veterinary health.

## 1. Introduction

Deer keds are hematophagous flies that frequently parasitize white-tailed deer [WTD; *Odocoileus virginianus* (Zimmermann, 1780)] and other cervids [1,2]. WTD are important reservoirs for tick-borne pathogens throughout their range [3,4] and serve as sentinels for mosquito-borne arboviruses [5,6]. WTD have also been implicated in the enzootic transmission of SARS-CoV-2 [7,8]. The intimate association of deer keds with WTD warrants further investigation of these ectoparasites to perpetuate pathogen transmission.

In the United States, there are four species of deer keds present in different geographical areas. *Lipoptena cervi* (Linnaeus, 1758) is present in the northeastern area of the United States, and Canada, *L. mazamae* (Rondani, 1878) is present in the southeastern area of the United States, and *L. depressa* (Say, 1823) and *Neolipoptena ferrisi* (Bequaert, 1935) are primarily clustered in the western part of the United States [9]. Unlike other more common ectoparasites of WTD, such as *Ixodes scapularis* (Say, 1821) and *Amblyomma americanum* (Linnaeus, 1758) ticks, keds are considered a one-host ectoparasite. Winged deer keds search for a suitable host, shed their wings shortly after landing, and feed on one host for the remainder of their lifetime. After mating, a female ked will give birth to a pre-pupa, which immediately pupates and falls off the host. After completing metamorphosis, a winged adult will emerge to find a suitable host and continue the life cycle [1,2].

Deer keds can impact wildlife’s host behavior and health and occasionally bite humans. In Europe, large burdens of *L. cervi* have been implicated in the development of alopecia in moose [*Alces alces* (Linnaeus, 1758)] [10]. Reindeer [*Rangifer tarandus tarandus* (Linnaeus, 1758)] experimentally infested with *L. cervi* displayed significantly more restless behavior, such as shaking, scratching, and grooming, which may affect the general health of the parasitized animals [11]. In the United States, *L. mazamae* infestations have been suggested to cause more irritation and subsequent grooming behavior in WTD compared to ticks, especially among juvenile deer [12]. While wild animals are the preferred host, deer keds sometimes land on humans and can inflict multiple bites [13]. Painful skin lesions and dermatitis can result from the bites, with symptoms potentially lasting for months [13,14,15,16]. A recent study also reported that common commercial repellents are not effective against deer keds, highlighting the vulnerability of humans to deer ked bites [17].

A limited number of studies report the screening of pathogens in deer keds in the northeastern area of the United States. In Massachusetts, a small number of *L. cervi* (5/6) tested positive for *Bartonella schoenbuchensis* (Dehio, 2001), a Gram-negative bacterium that can be transmitted vertically in deer keds. *B. schoenbuchensis* has been suggested to play a role in the development of dermatitis in humans [18,19,20]. Tick-borne pathogens have also been detected in deer keds. *Borrelia burgdorferi* (Johnson, 1984) and *Anaplasma phagocytophilum* (Foggie, 1949), the causative agents of Lyme disease and anaplasmosis, respectively, were detected in deer keds collected in Pennsylvania and Maryland [21,22]. Despite their co-occurrence on cervid hosts, little is known about the competence of deer keds as vectors for pathogens typically found in ticks. In the present study, we surveyed deer keds throughout Massachusetts for the presence of common tick-borne pathogens.

## 2. Materials and Methods

Deer keds were collected from hunter-harvested WTD during the Massachusetts 2023 deer hunting season. The WTD were sampled throughout Massachusetts at deer check stations operated by the Division of Fisheries and Wildlife (MassWildlife). Samples were also collected from WTD at a venison processor and at a community-shared cooler used for the temporary storage of WTD prior to processing. Due to the time constraints and feasibility of sampling mainly at check stations where the primary focus was collecting WTD demographic and morphometric data, a targeted search of the head and neck area of WTD was conducted. This body area of WTD has been shown previously to support high densities of ectoparasites, including deer keds [23,24]. Searches were performed for a minimum of 2 min, though some deer could be examined for longer periods. The deer keds were removed from WTD using forceps, placed in plastic vials that permitted airflow, and stored at 4 °C until DNA extraction. The ectoparasites were visually identified prior to molecular analyses as deer keds and typed to species by morphological characteristics following the method of Skvarla and Machtinger [9].

Total nucleic acids were extracted from the samples using the Masterpure Complete DNA and RNA Purification Kit (LGC Biosearch Technologies, Madison, WI, USA) following the manufacturer’s recommendations. Briefly, each sample was added to a 2 mL tube with a metal bead and homogenized in a TissueLyser for 1.5 min at 23 Hz. After homogenization, the samples were incubated (65 °C for 15 min) with the lysis solution, and then proteins were precipitated with the MPC Protein Precipitation Reagent. The supernatants were transferred to new tubes, and nucleic acids were precipitated with isopropanol. The samples were washed with ethanol and resuspended in 60 ul of molecular-grade water. After nucleic acid extraction, each sample was tested by real-time PCR in two multiplex reactions to detect *B. burgdorferi*, *B. miyamotoi* (Fukunaga, 1995), *B. mayonii* (Pritt, 2016), *Babesia microti*, *Ehrlichia muris eauclairensis* (Pritt, 2017), and *A. phagocytophilum* as previously described [25,26]. The probes and primers used for pathogen detection are listed in Appendix A.

## 3. Results

### 3.1. Sample Collection

In total, 99 deer keds were collected from 288 WTD at 10 locations throughout Massachusetts (Table 1 and Appendix A). All keds were identified as *L. cervi*, which is the only previously reported deer ked species present in Massachusetts [9]. The sampling locations comprised nine towns and 7 of the 14 Massachusetts counties. At least one ked was collected from each sampling location. More than half of the keds (52/99) were collected from the community-shared cooler located at West Tisbury on the island of Martha’s Vineyard, probably due to the lack of time constraints imposed on collections at this location.

### 3.2. Pathogen Screening

*A. phagocytophilum* was the only pathogen detected in the keds with an infection prevalence of 30% [(30/99), (95% confidence interval 0.21–0.40)] (Table 2). No other pathogen DNA was detected in the deer keds.

## 4. Discussion

Deer keds were collected from hunter-harvested WTD in Massachusetts to gain baseline knowledge of their distribution and pathogen prevalence. Deer keds were collected from each of the 10 geographically separated sampling sites, with 18.1% of the WTD having one or more keds. Previous studies have recorded deer keds collected from Dukes, Hampshire, and Worcester counties in Massachusetts [9,18]. In our study, we collected deer keds from those counties in addition to Berkshire, Hampden, Middlesex, and Plymouth counties. We expect that more deer keds will be found in other previously unrecorded areas in Massachusetts and neighboring states if more sampling efforts are performed.

The observed pattern of pathogen detection in the deer keds is likely a result of the deer ked life cycle and the reservoir competence of WTD for the tested pathogens. Since deer keds are one-host ectoparasites that primarily parasitize cervids, any pathogen detected in the deer keds was likely transmitted from the host during blood feeding. Co-feeding may be another route of pathogen transmission since deer keds and ticks feed simultaneously on WTD. However, the frequency of pathogen transmission by co-feeding appears to be very low [22]. The only pathogen DNA detected in the WTD-collected keds was *A. phagocytophilum*. This infection was likely acquired during blood-feeding since WTD are known reservoirs for certain variants of *A. phagocytophilum* [27,28]. We expect the *A. phagocytophilum* infection in deer keds to be the WTD-associated variant, but our molecular assay could not make this distinction. In contrast, WTD are not competent reservoirs for *B. burgdorferi* due to the highly efficient killing of spirochetes by WTD serum [29,30]. This serum-mediated killing explains why no deer keds were positive for *B. burgdorferi*, even though there is a high prevalence of *B burgdorferi* infection in *I. scapularis* ticks in Massachusetts [31]. These results contrast with a previous study that detected *B. burgdorferi* in deer keds from Pennsylvania [21]. It is unclear why those deer keds were infected with *B. burgdorferi* since they were also collected from hunter-harvested WTD. We did not find any deer keds infected with *B. miyamotoi* in our collection, although some evidence suggests that WTD may be a competent reservoir for *B. miyamotoi* [32]. *B. miyamotoi* is a low-prevalence pathogen in questing ticks, and the general prevalence of infection in WTD in the northeastern area of the United States is unknown [33]. Further work is needed to determine how often keds feed on *B. miyamotoi*-infected deer and if the infection can be passed on to deer keds. For the other tested pathogens, WTD were either not competent reservoirs (*B. microti*) or the pathogens were not expected to be present in our study area (*E. muris eauclairensis*) and *B. mayonii*) [34,35].

The detection of *A. phagocytophilum* DNA in deer keds does not indicate these insects as vector-competent. The determination of this would require carefully designed studies to determine if deer keds can transmit *A. phagocytophilum* to susceptible hosts. Even if deer keds can transmit the bacterium, it would likely be the deer-associated variant(s) of *A. phagocytophilum*, which are thought to be nonpathogenic to humans [27,28,36]. For these reasons, the risk of a deer ked bite causing anaplasmosis in humans is likely low.

Determining whether *A. phagocytophilum* can be transovarially transmitted could help elucidate the biological significance of pathogen DNA in deer-collected keds. This would require the pathogen testing of winged keds that have not yet fed on a host. A previous study in Hungary found that a low percentage (2%) of winged deer keds were positive for *A. phagocytophilum* [19]. Deer keds may pose little medical threat to humans, but if *A. phagocytophilum* can be transmitted vertically, deer keds may be important in the transmission cycle of certain *A. phagocytophilum* variants. A similar idea was previously suggested for another one-host ectoparasite of cervids, *Dermacentor albipictus* [37]. It is also intriguing to speculate how keds might be used to deliver WTD-specific pathogens, such as a deer variant of *A. phagocytophilum,* which have been engineered to interfere with the transmission of human pathogens directly or by precluding other ectoparasites.

Deer keds have clear utility as sentinels for the detection of deer-associated microbes, including a number of pathogen species associated with ticks [38]. *I. scapularis* ticks commonly parasitize WTD as adults but often feed on other host species in preimaginal stages [39]. This life cycle makes it challenging to discern whether infection in deer-collected adult ticks originates from the deer or another prior host. Since deer keds are associated with only one host, they may be used to detect the presence of microbes present in WTD. WTD can be infected with *A. phagocytophilum*, *E. chaffeensis* (Anderson, 1992), *Plasmodium odocoilei*, *Babesia odocoilei*, *Trypanosoma cervi*, and *Theileria cervi* [27,40,41,42,43,44]. Many of these pathogens are understudied. Sampling deer keds could augment surveys to inform the prevalence, genetic diversity, and distribution of these microbes. Although it has been reported that deer keds may switch hosts during the host breeding season [2,45], the frequency of this occurring in WTD is unknown, and keds may still be used to survey pathogens in host populations even if they sometimes move among individuals.

The role of deer keds as vectors in enzootic pathogen transmission needs to be explored further. Given the occasional spillover of ectoparasites to opportunistically bite humans or companion animals, these further studies may have public health and/or veterinary significance.

## Figures and Tables

**Table 1 insects-16-00042-t001:** Deer keds collected from WTD throughout Massachusetts.

County	Town	N Deer (with Keds)	N Keds
Berkshire	Dalton	23 (1)	4
Dukes	West Tisbury ^2^	53 (22)	52
Dukes	Edgartown	22 (4)	5
Hampden	Palmer	29 (8)	12
Hampshire	Belchertown ^1^	25 (7)	7
Hampshire	Belchertown	28 (3)	4
Middlesex	Ayer	30 (1)	1
Middlesex	Hopkinton	33 (3)	4
Plymouth	Middleborough	23 (2)	9
Worcester	Webster	22 (1)	1
	Totals	288 (52)	99

^1^ Venison processor. ^2^ Community-shared cooler.

**Table 2 insects-16-00042-t002:** Pathogen prevalence in deer keds.

County	Town	No. of Positive Keds/Total ^3^					
		Ap	Bm	Bb	Bmi	Bma	Em
Berkshire	Dalton	0/4	0/4	0/4	0/4	0/4	0/4
Hampshire	Belchertown ^1^	3/7	0/7	0/7	0/7	0/7	0/7
Hampshire	Belchertown	2/4	0/4	0/4	0/4	0/4	0/4
Hampden	Palmer	2/12	0/12	0/12	0/12	0/12	0/12
Worcester	Webster	1/1	0/1	0/1	0/1	0/1	0/1
Middlesex	Ayer	0/1	0/1	0/1	0/1	0/1	0/1
Middlesex	Hopkinton	0/4	0/4	0/4	0/4	0/4	0/4
Plymouth	Middleborough	0/9	0/9	0/9	0/9	0/9	0/9
Dukes	West Tisbury ^2^	20/52	0/52	0/52	0/52	0/52	0/52
Dukes	Edgartown	2/5	0/5	0/5	0/5	0/5	0/5
	Totals	30/99	0/99	0/99	0/99	0/99	0/99

^1^ Meat processor. ^2^ Community-shared cooler. ^3^ Ap = A. phagocytophilum; Bm = B. microti; Bb = B. burgdorferi; Bmi = B. miyamotoi; Bma = B. mayonii; Em = Ehrlichia muris eauclairensis.

## Data Availability

Raw data supporting the conclusions of this article will be made available by the authors on request.

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
