# Peer review of "Detection of Anaplasma phagocytophilum DNA in Deer Keds: Massachusetts, USA"

_insects, 2025, doi:10.3390/insects16010042_

Round 1
Reviewer 1 Report
Comments and Suggestions for Authors
There should be morphological or molecular identification od the deer keds. If deer keds have single host then what is its significance if it is not transferring the disease to other deer. The sample size and methods also are too general and does not present a good sample size.
Author Response
Response to Reviewer 1 comments.
We thank the reviewer for taking the time to review our manuscript. Please find our responses (in red text) to the reviewer comments (in black text) below.
Comments and suggestions for authors.
Comment 1: There should be morphological or molecular identification of the deer keds.
Response 1: Keds were identified using the key of Skvarla and Machtinger (2019). All individuals were identified as L. cervi, consistent with the known geographic range. Dorsal and ventral micrographs of each specimen are provided in supplemental materials.
Comment 2: If deer keds have single host then what is its significance if it is not transferring the disease to other deer.
Response 2: Vertical transmission of A. phagocytophilum (or other pathogens) among deer keds could lead to transmission of A. phagocytophilum variants from one deer to another (ie. infection acquired in first blood meal, then maintained transovarially and transmitted to a future host.
Comment 3: The sample size and methods also are too general and does not present a good sample size.
Response 3: Sample sizes reflect the timed sampling effort from hunter-harvested white-tailed deer during 2023. Similar published studies on keds in the northeast United States have had smaller sample sizes. Please see references below which are included in our manuscript.
Matsumoto, K.; Berrada, Z.L.; Klinger, E.; Goethert, H.K.; Telford, Iii, S.R. Molecular Detection of Bartonella Schoenbuchensis from Ectoparasites of Deer in Massachusetts. Vector-Borne and Zoonotic Diseases 2008, 8, 549–554, doi:10.1089/vbz.2007.0244.
Buss, M.; Case, L.; Kearney, B.; Coleman, C.; Henning, J. d. Detection of Lyme Disease and Anaplasmosis Pathogens via PCR in Pennsylvania Deer Ked. Journal of Vector Ecology 2016, 41, 292–294, doi:10.1111/jvec.12225.
Reviewer 2 Report
Comments and Suggestions for Authors
It is a well-written manuscript - I only have a few formal suggestions - marked up in the attached pdf

Author Response
Response to Reviewer 2 comments.
We thank the reviewer for taking the time to review our manuscript. Please find our responses (in red text) to the reviewer comments (in black text) below.
Comments and suggestions for authors.
Comment 1: It is a well-written manuscript - I only have a few formal suggestions - marked up in the attached pdf
Response 1: below with annotation where appropriate.
- Removed italics from “spp” in abstract.
- We have added the original author and year when Latin names are first introduced.
- Deleted “adult” from line 56.
- Added another reference (below) to the introduction in line 71.
- Oboňa, J.; Fogašová, K.; Krišovský, P.; Mlynárová, L.; Sikora, B.; Hromada, M. The First Report of Human Biting by Lipoptena Cervi from Slovakia. Annals of Parasitology 2024, 70, 47–49, doi:10.17420/ap7001.516.
- Italicized “B” in schoenbuchensis in line 76.
- Added a sentence to the discussion 191-194 acknowledging that keds may move among hosts but can still be used as sentinels to detect pathogens in host populations.
- Updated Figure S1 map to have scale and north pointer.
Reviewer 3 Report
Comments and Suggestions for Authors
This is a properly planned and executed study, the results of which deserve to be published. What makes me wonder is why it has not been sent to a more topic-appropriate journal, such as Pathogens.
I have a few comments that could improve this manuscript:
1. Keywords should not repeat terms that are already included in the title; it is worth here to include the Latin names of the parasites tested and their hosts and other words relevant to this study.
2. Line 44 and elsewhere in the text: sentences should not begin with abbreviations, including abbreviated Latin names.
3. L85: what it is “MassWildlife”? Any reference?
4. L93-96: identification of ectoparasitic species is unclear: it cannot be done on the information that others species are not present in the area (in this way others will never be recognized). The authors should refer to a species-identification key with recognizable character traits.
5. L97-107: DNA isolation is not as relevant to this study as the methodology for pathogen DNA detection. Therefore, either instead of or in addition to, information on which species/groups of pathogens were screened for and how they were identified should be included. The reference here to the two publications of the co-author is inappropriate.
6. One location (West Tisbury, community shared cooler) is responsible for 2/3 of Anaplasma hits. This should be better explained/discussed in the Discussion section.
Author Response
Response to Reviewer 3 comments.
We thank the reviewer for taking the time to review our manuscript. Please find our responses (in red text) to the reviewer comments (in black text) below.
Comments and suggestions for authors.
This is a properly planned and executed study, the results of which deserve to be published. What makes me wonder is why it has not been sent to a more topic-appropriate journal, such as Pathogens.
I have a few comments that could improve this manuscript:
Comment 1. Keywords should not repeat terms that are already included in the title; it is worth here to include the Latin names of the parasites tested and their hosts and other words relevant to this study.
Response 1: Thank you and duly noted.
Comment 2. Line 44 and elsewhere in the text: sentences should not begin with abbreviations, including abbreviated Latin names.
Response 2: corrected.
Comment 3. L85: what it is “MassWildlife”? Any reference?
Response 3: We have added this clarification to the manuscript (line 87).
Comment 4. L93-96: identification of ectoparasitic species is unclear: it cannot be done on the information that others species are not present in the area (in this way others will never be recognized). The authors should refer to a species-identification key with recognizable character traits.
Response 4: Keds were identified using the key of Skvarla and Machtinger (2019). All individuals were identified as L. cervi, consistent with the known geographic range. Dorsal and ventral micrographs of each specimen are provided in supplemental materials.
Comment 5. L97-107: DNA isolation is not as relevant to this study as the methodology for pathogen DNA detection. Therefore, either instead of or in addition to, information on which species/groups of pathogens were screened for and how they were identified should be included. The reference here to the two publications of the co-author is inappropriate.
Response 5: To better describe the real-time PCR assay, we have added a supplemental table (Table S1) that lists the primers and probes used to detect the pathogens. We have also added a sentence to the end of the methods section (line 110).
Comment 6. One location (West Tisbury, community shared cooler) is responsible for 2/3 of Anaplasma hits. This should be better explained/discussed in the Discussion section.
Response 6: The West Tisbury yielded more keds because the collector at this location had more time to search the white-tailed deer for keds. Most collections at other sites were restricted to the two minute period while hunters waited. It is true that the West Tisbury site had 2/3 of the Anaplasma infected keds, but it also have more than ½ of the total keds in the study.
Round 2
Reviewer 1 Report
Comments and Suggestions for Authors
both previous objections answered and relevant procedures and references added.